# Conjugated linoleic acid as a novel insecticide targeting the agricultural pest *Leptinotarsa decemlineata*

**Justin Clements**[1], **Russell L. Groves**[1], **JoAnn Cava**[1], **Caroline C. Barry**[2], **Scott Chapman**[1], **Jake M. Olson**[2]*

**1** Department of Entomology, University of Wisconsin-Madison, Madison, Wisconsin, United States of America, **2** Department of Animal Sciences, University of Wisconsin-Madison, Madison, Wisconsin, United States of America

* jmolson22@wisc.edu

**Data Availability Statement:** All relevant data are within the paper and its Supporting Information files.

## Abstract

The Colorado Potato Beetle, *Leptinotarsa decemlineata*, is a major agricultural pest of solanaceous crops in the United States. Historically, a multitude of insecticides have been used to control problematic populations. Due to increasing resistance to insecticides, novel compounds and methodologies are warranted for the control of beetle populations. Mixed-isomer conjugated linoleic acid has been studied in-depth for its beneficial properties to mammalian systems. At the same time, studies have demonstrated that conjugated linoleic acid can manipulate fatty acid composition in non-mammalian systems, resulting in embryo mortality. Consequently, experiments were conducted to assess the effects of foliar-applied conjugated linoleic acid on larval growth, embryogenesis, and feeding preference in Colorado potato beetle. Both maternal and deterrent effects of dietary conjugated linoleic acid were assessed. Conjugated linoleic acid demonstrated desirable insecticidal properties, including increased larval mortality, slowed larval development, antifeedant effects, and decreased egg viability after maternal ingestion.

## Introduction

The Colorado potato beetle, *Leptinotarsa decemlineata* (Say) is a major agricultural pest of both commercial and subsistence solanaceous crops. With a range encompassing North America, Europe, and Asia, it is considered one of the greatest insect threats to agriculture and food security due to its ability to develop pesticide resistance at alarming rates [1]. If not properly controlled, yield reduction of between 50–100% can be observed in solanaceous crops, including commercial potato cultivars which have a production value of approximately $4 billion annually (United States, 2016) [2]. Historically, L. decemlineata populations have been controlled with numerous insecticidal compounds including, but not limited to, DDT, paris green, arsenical pesticides, and other historical insecticides [1]. Current management includes combinations of insecticides that are both foliar and soil-applied, and systemic chemistries from multiple insecticide resistance action committee (IRAC) modes of action (MoA) groups.

**Funding:** The support for this research was provided by the University of Wisconsin - Madison Office of the Vice Chancellor for Research and Graduate Education with funding from the Wisconsin Alumni Research Foundation to JO and RG. The funder had no role in study design, data collection and analysis, decision to publish, or preparation of the manuscript.

**Competing interests:** The authors have declared that no competing interests exist.

One of the most common insecticidal chemistries currently used to control beetle populations are the neonicotinoids, an IRAC Group 4A MoA [3]. Since their introduction in the mid-1990s, beetle populations have developed noticeable resistance; however, neonicotinoids are still one of the primary tools used for crop protection targeting *L. decemlineata* [4]. In recent years, this class of insecticides has undergone strong scrutiny by the public and agricultural community for off-target effects [5,6]. Subsequently, the agricultural community is searching for alternatives to current insecticide regiments.

Several biorational and reduced-risk insecticidal compounds are currently being investigated as safer alternatives for insecticidal control, including synthetic nucleic acids (RNA interference) [7], bacterial and fungal secondary metabolites (macrocyprins) [8], and even microbial fermentation products (spinosad). Compounds capable of disrupting lipid metabolism offer insecticidal properties which would be useful for the control of *L. decemlineata* [9]. Mixed-isomer conjugated linoleic acid (CLA) has been intensely studied for promoting human and animal health and is most widely recognized to affect carcinogenesis, atherosclerosis, inflammation, immunity, metabolic syndrome, bone mass, and lipogenesis [10–19]. CLA has been previously explored for its insecticidal properties on the European corn borer (*Ostrinia nubilalis*) and the common housefly (*Musca domestica*) with mixed results. A study conducted with European corn borer showed that a dietary intake of up to 0.6% CLA resulted in decreased $\Delta^9$-desaturase activity in eggs, pupae, and adults with an associated decrease in larval growth and survival [20]. Egg embryogenesis was not impacted by maternal dietary CLA, which was previously reported in laying hens [21, 22]. Park et al., 2006 observed that silkworms fed on a 10% CLA diet had significantly reduced body weight and food intake compared to control silkworms over the 4th instar stage [23]. In the common house fly, researchers concluded intake up to 10% CLA had no adverse fitness effects on any stage of the insects evaluated [24]. These studies suggest that the effects of CLA on specific insects may be taxa dependent.

In the current study, our primary goal was to assess the insecticidal properties (preventing, destroying, repelling, or mitigating pests) [25] of mixed-isomer CLA in *L. decemlineata*. Coleoptera comprise many agriculturally-relevant pests and are an appropriate and relevant order to explore the effects of CLA for insecticidal properties. The effects of CLA on Coleopteran species have not been previously evaluated. Based on the prior literature, we hypothesized that the consumption of CLA by gravid, adult female beetles would result in incorporation of CLA into egg lipids, resulting in adverse hatch rates. Additionally, we investigated whether CLA would have antifeedant effects on early larva, resulting in the potential use of CLA as a dual mode of action insecticide. Using laboratory dose-response studies, lipid analysis, and a greenhouse-based spray evaluation, we established that CLA has the potential to be an effective biorational insecticide.

## Methods

### Ethical approval

This article does not contain studies with any human participants and no specific permits were required for field collection or experimental treatment of *L. decemlineata* for the study described.

### Rearing beetles and potatoes

Approximately 300 adult beetles were initially collected on June 20th, 2016 from the Arlington Agricultural Research Station, Arlington, Wisconsin (AARS, 43.315527, -89.334545). Previous studies have shown this population remains highly susceptible to insecticides [26, 27]. Adult

beetles were hand-collected from the canopy of potato plants, placed in plastic containers, and transported to the University of Wisconsin-Madison. Reproducing populations of *L. decemlineata* were sustained on healthy potato plants in mesh cages under a 16:8 hour light:dark (L: D) photoperiod. *Solanum tuberosum* (potatoes) cultivar Russet Burbank were grown from seed tubers at the University of Wisconsin-Madison, and no chemical compounds were used in growing potatoes. Untreated foliage from potato plants was obtained from plants grown at the University of Wisconsin-Madison greenhouse and provided to beetles daily. Adult beetles were given the opportunity to randomly mate and oviposit on potato foliage. Egg masses were collected daily and placed on filter paper in 100 x 15 mm petri dishes (Corning, Corning, New York) and held at 26˚C, 70% relative humidity (RH), and 16:8 (L:D) photoperiod. Following egg hatch, larvae were provided untreated foliage daily and maintained as cohort groups throughout the remainder of their larval development. Prior to pupation, larvae were transferred to mesh cages with vegetative potato plants and maintained throughout emergence as adults.

## General procedures for Larval Feeding Bioassays

The CLA used for each bioassay was a 60% mixed-isomer preparation comprised of a 50:50 mixture of the cis-9, trans-11 and trans-10, cis-12 isomers (BASF, Germany). Impurities included 5% palmitate (16:0), 3% stearate (18:0), 25% oleate (18:1c9), 1% vaccenate (18:1c11), 2% linoleate (18:2n-6), and 4% unknowns.

From the previously described lab colony, second instar larvae were identified according to Boiteau et al. [28]. Individual larvae were placed in individual wells of a 12-well Falcon (Corning Inc., Corning, New York) culture dish. Each well contained a water-dampened sponge, covered by filter paper giving the larva a platform on which to stand and feed. Every 24 hours, larvae were presented with new treatments of potato leaf disks (2.016 cm$^2$). To track the progress of larval development throughout each assay, each larva was weighed using an AE 100 analytical balance (Mettler Toledo, Columbus, OH) every 24 hours. Larva weight gain was recorded for only surviving insects. Data were statistically analyzed with a ANOVA with a Tukey, post-hoc analysis to determine significant changes in weight gain with $p \leq 0.05$ considered significant.

## Larval Feeding Bioassay (CLA-acetone carrier)

A sub-sample of 50, 2nd-instar larvae (n = 10 per treatment group) from the lab colony were distributed in culture dishes as described in the previous section. Leaf disks were dipped and then immediately removed from experimental solutions of acetone or in a solution of CLA (2%, 4%, 8% and 16% v/v in acetone). Dipped leaves were allowed to dry for 10 minutes in a chemical fume hood before they were weighed and subsequently placed in larval microenvironments. The average amount of CLA on foliage after leaf dips was 0, 61, 103, 168, and 335 mg/g leaf disc (SD ±10 mg) for the 0 (i.e. acetone), 2, 4, 8 and 16% CLA leaf dips, respectively. The above-stated amounts corresponded to absolute levels of 1.96, 1.99, 3.75, and 7.50 mg CLA (SD ± 0.6 mg) per leaf disc (average weight of 20mg; SD ±3mg per leaf disc). Larvae were weighed at the start of the bioassay and every 24 hours until trial end. The average change in weight served as the indicator of the effectiveness of the treatment, as well as any observed mortality and growth irregularities.

## Larval Feeding Bioassay (CLA-aqueous suspension)

A subsample of 50, 2nd-instar larvae (n = 10 per treatment group) from the lab colony were distributed in culture dishes as described in the previous section. Leaf disks were briefly dipped

into an oil-in-water emulsion comprised of 0.125% v/v polyoxyethylene (20) sorbitan monolaurate (Tween-20, Sigma Aldrich; carrier) in $H_2O$ (control) or 0.125% v/v Tween-20 in $H_2O$ + CLA (2, 4, 8, or 16% v/v) and dried in a fume hood for 10 minutes. Larvae were fed as previously described and weights were recorded every 24 hours over a 96-hour trial period.

## Greenhouse-based spray evaluations

*Leptinotarsa decemlineata* larvae (n = 20, 10 per treatment group) were placed in culture dishes as previously described. Six mature potato plants from the University of Wisconsin-Madison greenhouse were sprayed with either 16% CLA emulsified in 0.125% aq. tween-20 in water or a water control once at the beginning of the study. Spray was delivered at 20 gallons per acre with a twin, flat-fan, double nozzle boom. Mature, untreated plant leaves were used as a control. Treated plants were stored separately from untreated potato plants under optimal growing conditions. Leaf disks (2.016 $cm^2$) were cut from the mature, outer canopy leaves and fed to larvae every 24 hours. Larval weights were recorded daily over the course of a 96-hour trial period as previously described. Although not quantified, the primary CLA isomers (c9t11 and t10c12) were detected on sprayed foliage using GC-FID analysis through 120 hours post-spraying, while CLAs were not detectable in control foliage at any time.

## Adult feeding assays

To investigate the effects of maternally fed CLA on oviposition, hatch rate, and larval survival, 16 adult beetles (8 females, 8 males) of the same age and fitness were selected from the previously described lab colony and assigned to dietary treatment groups. Beetles were fed leaf material (approx. 0.25 grams) dosed with 30 ul of either 16% CLA daily or a no-treatment control. After a one-week treatment period, pairs of beetles from the same treatment groups were allowed to mate for 24 hours before they were separated back into their respective petri dishes. Females were observed for the oviposition of egg masses until approximately 10 egg masses were collected from each dietary treatment group. During this time the female beetles were dosed with their treatments until completion of the assay. These egg masses were removed daily and put into separate labeled petri dishes on filter paper. They were counted and observed for hatch rates throughout the assay. Two-way student's T-tests were conducted to compare hatch rates in control and CLA feed insects.

## Tissue fat extraction and fatty acid composition analysis

Total fat content of samples (foliage, 1g; eggs, 0.2g) was extracted in Folch reagent as previously reported by Folch et al. 1957, using dichloromethane as a substitute for chloroform [29]. Specifically, for foliage samples, isopropyl alcohol (3ml) was added prior to tissue homogenization to deactivate lipases. Total extracted fatty acids were methylated using 0.5M sodium methoxide similar to methods described by Christie [30] using select modifications described by Politz et al. [31]. Briefly, toluene was added to dried dichloromethane extract (2:1 v/w). Next, 0.5M sodium methoxide was added in excess to lipid extracts (100:1 v/w) and samples were heated at 60˚C for 10 minutes in a water bath. The methylation reaction was quenched with 0.35M glacial acetic acid (1.5:1 v/v) followed by hexane extraction of methyl esters to yield a final FAME concentration of 10mg/ml. Relative abundance of fatty acid methyl esters (FAME) was analyzed using gas chromatography (Agilent 6890N) coupled with flame ionization detection (GC-FID) as previously described [32]. A 100m biscyanopropyl polysiloxane capillary column (Rt-2560, Restek Corp, Bellefonte, PA) was used for separation of FAMEs. FAME were identified using a custom qualitative FAME standard (Matreya LLC, Pleasant Gap, PA,

#SPL4833). Two-way student's T-tests were conducted to compare means of individual fatty acids between groups.

### Adult choice assay

From the previously described lab colony, 10 adult female beetles were identified according to Boiteau et al. [28]. Each individual was placed in a petri dish and given two leaf disks (2.016 cm$^2$). Disks were dipped in either acetone/tween (control) or a 16% v/v concentration of CLA/tween/acetone and immediately removed. The leaf disks were visually assigned a percentage of leaf disk consumed by the beetle after 24 hours. This procedure was repeated over 5 days. The average percentage of leaf disk consumed was computed and compared for both groups. Two-way student's T-tests were conducted to compare choice assay.

### Larval choice assay

From the previously described lab colony, 30 second instar larvae (n = 10 insects per treatment group) were identified according to Boiteau et al. [28]. Each individual was placed in a petri dish and given a choice between two leaf disks (2.016 cm2); one control (acetone dipped) leaf disk and one leaf disk dipped in one of the following treatments: 1) a 16% v/v concentration of CLA in acetone, 2) a 16% v/v c9t11 methyl ester in acetone, or 4) a 16% v/v t10c12 methyl ester in acetone. The c9t11- and t10c12-CLA isomers were >98% pure preparations as determined by GC-FID analysis (Matreya, State College, PA). The leaf disks were visually assigned a percentage of leaf disk consumed by the larvae after 24 hours. The average percentage of leaf disk consumed was computed and compared for both groups. Two-way student's T-tests were conducted to compare choice assay.

## Results

### Larval Feeding Bioassay (CLA-acetone carrier)

Conjugated linoleic acid affects the growth of 2$^{nd}$ instar *L. decemlineata* larva in a dose dependent manner. Larvae fed potato foliage dosed with 2, 4, 8 or 16% CLA (v/v) resulted in 30, 80, 100, and 100% mortality, respectively, within 96-hours (Fig 1A). Larvae fed doses greater than 2% showed either significantly reduced weight gain or no weight gain compared to controls (Fig 1B). Larvae fed 2% CLA-dosed foliage displayed no significant weight differences from control larvae (Fig 1B). Negative weight gains in treatment groups were observed, demonstrating weight loss from the start of the study.

### Larval Feeding Bioassay (CLA-aqueous carrier)

Larval growth and survival in response to the foliar-applied CLA treatments (aqueous suspension; 2, 4, 8, 16% CLA v/v) were similar to the outcomes of the acetone-carrier CLA assay. Mortality of 20, 50, 80, and 100% was observed in larvae fed 2, 4, 8, and 16% CLA treatments, respectively, over 96 hours (Fig 2A). Larvae fed most doses showed significantly reduced weight gain or no weight gain compared to controls at different time points throughout the assay (Fig 2B). Negative weight gains in treatment groups were observed, demonstrating weight loss from the start of the study.

### Greenhouse-based spray evaluations

Based on pilot data, a twin, flat-fan nozzle body (TeeJet XVR80–006) affixed to a dual-nozzle boom provided the best spray coverage of potato foliage when delivered at an application volume of 20-gal/acre (S1 File). Plants were sprayed only once and the resulting treated-foliage

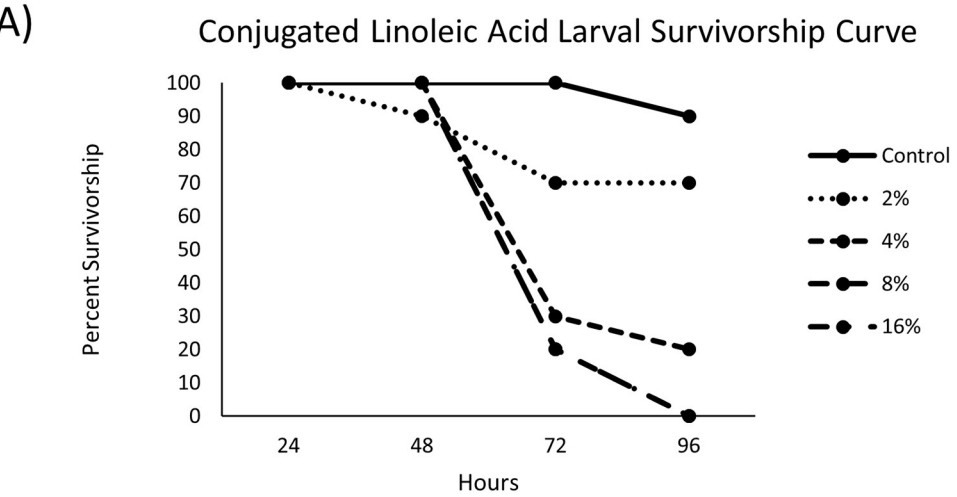

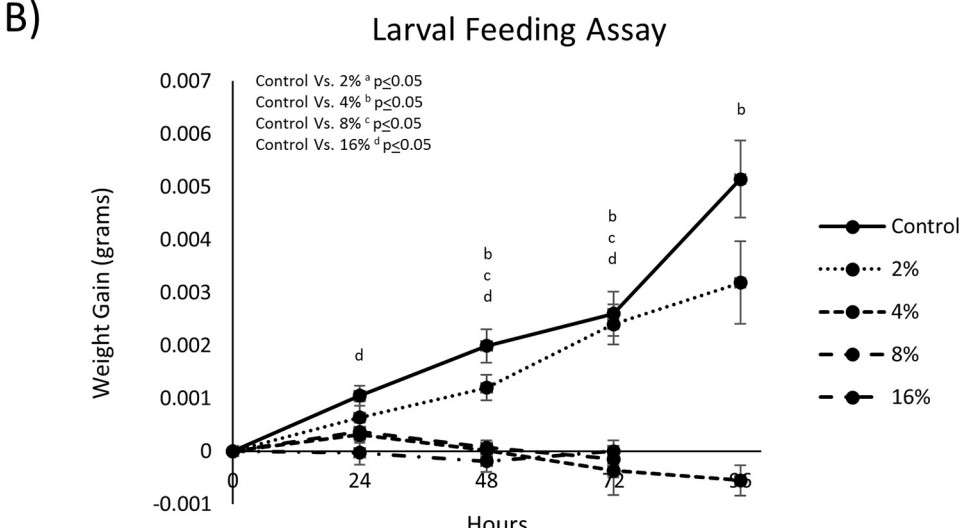

**Fig 1.** (A) Survivorship curves of 2$^{nd}$ instar *L. decemlineata* larvae following treatment with doses ranging from 2–16% foliar-applied CLA in acetone carrier over 96-hours (8% and 16% had the same survivorship curve). (B) 2$^{nd}$ Instar *L. decemlineata* larval growth rates during treatment with foliar-applied CLA. Doses beyond 2% CLA (v/v) significantly reduced larval growth over 96-hours (data represents mean weight gain ± S.E.).

was removed daily and fed to beetle larvae on consecutive days over a 5-day feeding bioassay. A cumulative mortality rate of 70% was observed in conjunction with a decreased weight gain in larvae fed foliage from the 16% CLA group compared to 30% mortality in the controls (Fig 3A). Decreased weight gain was observed in larvae fed 16% CLA-treated leaf material compared to the control (unsprayed foliage) (Fig 3B). Both CLA isomers (trans-10, cis-12 and cis-9, trans-11 CLAs) were detected on potato foliage of the 16% CLA treatment at 24- and 120-hours post-spray using GC-FID analysis.

## Adult feeding assay

Egg viability was significantly reduced in adult females fed foliage treated with 16% CLA over both short (~2 weeks) and long-term (~4 weeks) consumption period (Table 1). Egg viability, as measured by hatch rate, was 41% and 29% respectively, in the CLA-treated groups

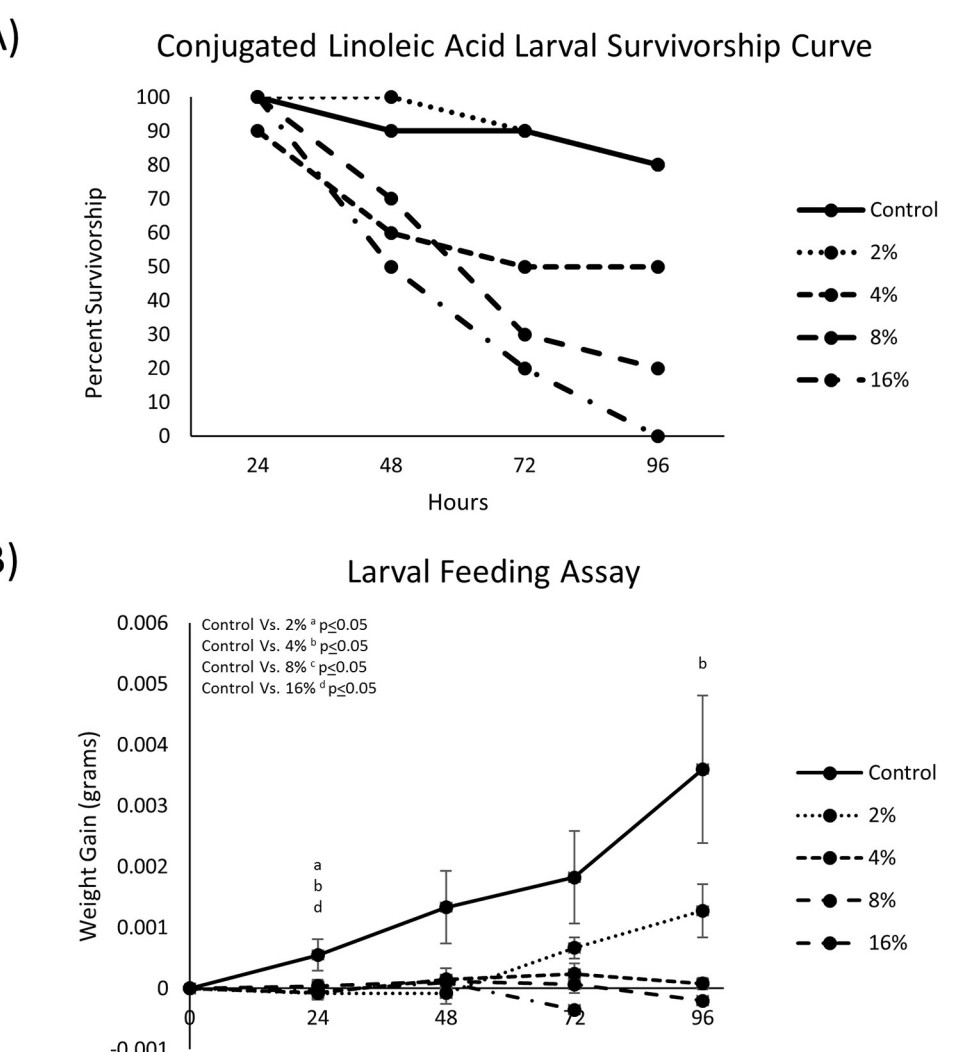

**Fig 2.** (A) Survivorship curves of 2nd instar *L. decemlineata* larvae after being fed 2–16% CLA-treated (aqueous suspension) foliage over 96-hours. (B) 2nd Instar *Leptinotarsa decemlineata* larvae weight gain after feeding of CLA-treated foliage over 96-hours (data represents mean weight gain ± S.E.).

compared to 100% and 94% in controls over both short and longer term CLA consumption intervals. There were no significant differences in egg viability between exposure times (i.e. short vs long) within each treatment. There was no significant difference in the amount of eggs laid between the control group and the treatment groups.

## Fatty acid composition analysis

Total fatty acid composition of *L. decemlineata* egg clutches showed several significant changes between 16% CLA-treated and controls (Table 2). Total saturated and mono-unsaturated fatty acids were decreased in the 16% CLA group compared to controls ($P = 0.028$ and $P < 0.001$, respectively). Polyunsaturated fatty acids were increased overall in the 16% CLA group compared to controls ($P < 0.001$). Individual CLA isomers (t10c12 and c9t11) were detected in the 16% CLA groups only. Significant changes in fatty acids within the 16% CLA group versus controls were manifest as specific reductions in 16:0, 20:0, 18:1c9, 18:1c11, 18:2n-6, 18:3n-6,

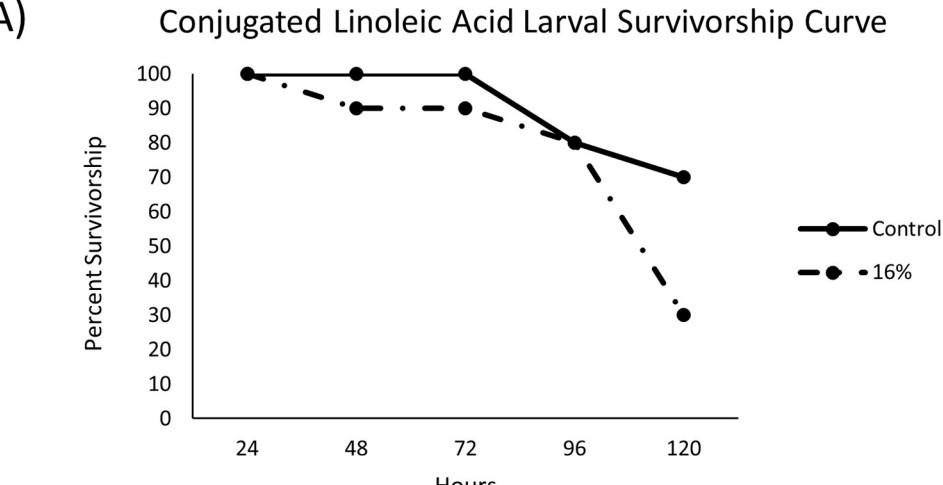

**Fig 3.** (A) Survivorship curves of 2nd instar *L. decemlineata* larvae after being fed a continual diet of 16% CLA foliage, treated using a controlled 20-gal/acre application volume. (B) 2nd instar *L. decemlineata* larvae weight gain illustrating a statistically significant decrease in weight gain over 120 hours compared to control (data represents mean weight gain ± S.E.).

and 18:3n-3, and an increase in CLA isomers. The Δ9-desaturase activity in eggs was decreased in the 16% CLA treated group compared to controls (*P = 0.027*).

## Adult choice assay

Adult beetles did not display a preference between leaf material treated with the control aqueous suspension comprised of water/tween-20 (0.125% v/v) and untreated leaf material

**Table 1. Adult *L. decemlineata* feeding assay and corresponding egg viability.**

| Treatment | N | Exposure | # Hatched | Mean percent Hatch ± S.E | Control Vs. Treatment P-value |
|---|---|---|---|---|---|
| Control | 145 | Short (2 weeks) | 145 | 100±0 | |
| Control | 182 | Long (4 weeks) | 174 | 94.44±3.69 | |
| 16% CLA | 120 | Short (2 weeks) | 51 | 40.71±14.69 | 0.002 |
| 16% CLA | 136 | Long (4 weeks) | 28 | 28.95±12.15 | 0.0007 |

**Table 2. Fatty acid composition in potato foliage and *L. decemlineata* eggs resulting from *adult feeding assay*.**

| Fatty acid | Potato foliage | Egg fatty acids | | P-value |
|---|---|---|---|---|
| | Untreated | Control | 16% CLA | |
| | % of total FAs | % of total FAs | | |
| 16:0 | 13.4 (0.14) | 6.14 (0.43) | 3.06 (0.37) | 0.002 |
| 17:0 | 0.09 (0.02) | 0.43 (0.05) | 0.26 (0.04) | 0.063 |
| 18:0 | 1.45 (0.07) | 14.4 (0.42) | 13.3 (0.60) | 0.188 |
| 19:0 | 0.08 (0.01) | 0.49 (0.03) | 0.26 (0.03) | 0.005 |
| 20:0 | 0.50 (0.03) | 1.87 (0.06) | 0.91 (0.09) | <0.001 |
| ∑SFA | 16.3 (0.33) | 23.6 (1.57) | 18.4 (1.03) | 0.028 |
| 18:1 c9 | 1.42 (0.04) | 24.8 (0.35) | 16.3 (0.53) | <0.001 |
| 18:1 c11 | 11.0 (0.29) | 2.64 (0.09) | 1.37 (0.20) | 0.002 |
| 20:1 c11 | 0.07 (0.01) | 0.38 (0.14) | 0.27 (0.04) | 0.428 |
| ∑MUFA | 12.7 (0.30) | 27.8 (0.33) | 18.1 (0.39) | <0.001 |
| 18:2 n-6 | 17.3 (0.17) | 20.9 (1.19) | 14.1 (0.93) | 0.004 |
| 18:3 n-3 | 46.7 (0.82) | 14.9 (0.50) | 5.35 (0.25) | <0.001 |
| 18:2 c9t11-CLA | 0.17 (0.02) | ND | 16.8 (0.56) | <0.001 |
| 18:2 t10c12-CLA | ND | ND | 16.0 (0.25) | <0.001 |
| 20:2 n-6 | ND | 2.28 (0.31) | 1.34 (0.35) | 0.087 |
| 20:3 n-3 | 0.09 (0.01) | 0.84 (0.20) | 0.12 (0.01) | 0.068 |
| 20:4 n-6 | 0.05 (0.01) | 0.66 (0.03) | 0.56 (0.09) | 0.385 |
| ∑PUFA | 64.6 (0.63) | 39.7 (0.76) | 54.1 (0.67) | <0.001 |
| Unknowns | 6.4 (0.11) | 9.1 (1.08) | 9.3 (1.89) | 0.901 |
| Total n-3 | 46.8 (0.82) | 15.7 (0.69) | 5.4 (0.28) | <0.001 |
| Total n-6 | 17.7 (0.17) | 23.9 (1.11) | 31.9 (1.03) | 0.002 |
| n-6:n-3 ratio | 0.4 (0.01) | 1.5 (0.13) | 6.0 (0.45) | <0.001 |
| $\Delta^9$-index[1] | NA | 0.55 (0.01) | 0.50 (0.01) | 0.027 |

Values are means ± (SEM). Individual leaf samples (n = 3) or egg masses (n = 3 per treatment) from different females were collected and analyzed. Data are pooled from two independent feedings trials.

[1]Calcualted as [18:1n9 / (18:1n9 + 16:0 + 18:0)].

(Table 3). In the treatment group given a choice between the untreated and 16% CLA (aqueous suspension)-treated leaf material, there was a trend toward decreased leaf consumption (-41%; *P = 0.1*) of treated foliage when compared to control.

## Larval choice assay

Larvae consumed significantly greater proportions of the untreated leaf materials (acetone treatment only) in pairwise comparisons with the aqueous suspensions comprised of the following treatments: 16% CLA v/v, 16% c9t11 v/v, and 16% t10c12 v/v (Table 4). The parent

**Table 3. Adult *L. decemlineata* choice assay of CLA applied to foliage as an aqueous suspension.**

| Treatment | Percent Consumed | Control Vs. Treatment P value |
|---|---|---|
| Control | 74.5±16.7 | |
| Aqueous control | 69.2±18.9 | P = 0.8 |
| Control | 72.7±15.9 | |
| 16% Aqueous CLA | 41±14.1 | P = 0.1 |

**Table 4. Larval *L. decemlineata* choice assay of CLA isomers.**

| Treatment | Percent Consumed | Control Vs. Treatment P value |
|-----------|------------------|-------------------------------|
| Acetone | 74±21.08 | |
| CLA | 16±15 | 8.70E-07 |
| Acetone | 60±32.01 | |
| c9t11-CLA | 17±17.63 | 0.0015 |
| Acetone | 79±22.23 | |
| t10c12-CLA | 18±20 | 3.2E-06 |

compound CLA and its two isomers all resulted in significant reductions in feeding activity of larvae over a 24-hour period suggesting both isomers in CLA could be feeding deterrents.

## Discussion

The Colorado potato beetle is a major agricultural pest of commercial potatoes. To date, populations of *L. decemlineata* have developed resistance to more than 56 different insecticidal chemistries [33]. As insecticide resistance continues to develop, growers face new challenges, including the need to control problematic populations. Further, current insecticides have been heavily scrutinized in recent years for off target effects, including the Group 4A MoA neonicotinoids. With increased resistance to commonly used insecticides and ecological concern for deleterious effects of the current insecticidal regiments, new biorational insecticidal compounds that are deemed as safer alternatives are needed for *L. decemlineata* control. The goal of this investigation was to determine whether CLA can be used as an effective biorational insecticide by disrupting the growth and survival of *L. decemlineata* through the utilization of laboratory bioassays and greenhouse-based spray evaluation.

Reducing the growth and survival at the 2nd instar stage of larval development prevents beetle maturation and subsequent defoliation of potato fields. A foliar application of CLA effectively reduced larval growth and survival at the 2nd instar stage, and the dose-specific effects of CLA, whether applied in an organic or aqueous solution, showed that the 16% CLA formulation caused 100% mortality over 4-days of dietary intake. Additional larval feeding trials with pure t10c12- and c9t11-CLA isomers demonstrate a decreased foliage intake similar in effect as the mixed-isomer CLA. These data may also suggest that the methyl ester moiety confers a general anti-feedant effect, as supported by data from Szafranek [34], although this requires further investigation.

To understand how CLA's effects would translate to an agriculture field, a greenhouse-spray experiment was performed and subsequently 2nd instar *L. decemlineata* larvae were fed CLA-treated leaf material. A 40% decrease in larval survival from control was observed in this trial; however, a decline in efficacy of 30% was observed when results were compared to the similar lab-based larval feeding bioassay (i.e. aqueous suspension assay). As many factors are involved in formulating active ingredients into sprayable insecticides, the decreased efficacy may have been due to possible non-homogeneity of CLA in the aqueous solution, unknown interactions between CLA and other formulation components (e.g. tween-20), and/or increased degradation of CLA in the aqueous solution. Future CLA insecticide formulations would benefit from an improved understanding of CLA's stability in suspension and post-application. It should, however, be stated that the t10c12- and c9t11-CLA isomers were detected on CLA-treated potato foliage throughout the entire greenhouse experiment, although amounts were not quantified.

Adult *L. decemlineata* showed a trend toward decreased feeding preference of foliage treated with 16% CLA in aqueous suspension. In the adults that did consume the 16% CLA-treated foliage (adult feeding assay), we discovered that CLA has a dual-modality and negatively affects *L. decemlineata* hatch rates in addition to post-hatch larval feeding deterrence. Egg hatch and larval survivability were significantly reduced in *L. decemlineata* adults fed 16% foliar-applied CLA. The fatty acid composition of eggs suggests that CLA incorporates broadly into egg lipids, as indicated by a significant reduction in total saturates, monounsaturates, and an increase in polyunsaturates. Alpha-linolenic acid (ALA; 18:3n-3) was decreased most significantly among the individual FAs (-64 percentage points versus controls).

CLA is an established inhibitor of stearoyl-CoA desaturase (aka. $\Delta^9$-desaturase) in fish, poultry, and mammals [35–37]. In this study, CLA caused a 9.1% decrease in $\Delta^9$-desaturase activity in egg lipids with an accompanying decrease in oleic acid (18:1c9). It is unclear whether the 9.1% inhibition of $\Delta^9$-desaturase caused the observed decrease in egg viability. Similarly, Gereszek [20] observed no deleterious effects on embryogenesis with an 18% decrease in $\Delta^9$-desaturase in the European corn borer. Park et al., 2006 observed that silkworms fed on a 10% CLA diet had significantly reduced body weight and food intake compared to control silkworms over the 4$^{th}$ instar stage, although no differences in $\Delta^9$-desaturase activity were observed [24]. Whether or not inhibition of $\Delta^9$-desaturase activity is effective for insect population management is likely taxa- and developmental stage-dependent and whether CLA acts primarily through this mechanism requires further investigation.

The reduction in egg lipid ALA due to CLA was unexpected based on review of literature involving the effects of CLA on embryogenesis [38–40]. Unlike some insects (i.e. silkworms), research from Cripps et al. [41] suggests that Coleopteran are not capable of *de novo* ALA or linoleic (LA; 18:2n-6) synthesis, and these fatty acids must be obtained from the diet. *L. decemlineata* egg lipid LA was also significantly decreased (-33%) due to CLA, suggesting the possibility that maternally-fed CLA may be capable of causing essential fatty acid deficiency in *L. decemlineata* eggs through displacement of ALA and LA. To our knowledge, the minimum requirements of ALA and LA have not been determined for *L. decemlineata*, or other Coleopterans. Kryzmanska [42], showed that fecundity of adult *L. decemlineata* is influenced by changes in both LA and ALA. Therefore, CLA may interfere with essential fatty acid incorporation and/or metabolism in eggs and may be a novel approach to controlling *L. decemlineata* populations.

New alternative and reduced-risk biorational insecticidal compounds are needed to reestablish control over agricultural resistant insect populations. Conjugated linoleic acid has been previously studied for its insecticidal properties against insects in the orders Diptera and Lepidoptera with limited success [20, 24]. Despite having little success in other insect orders, we have demonstrated an effective, dual-mode of action biorational insecticide option that very adequately controlled stages of *L. decemlineata*. Future work is needed to optimize CLA into an effective, field relevant insecticide including determining CLA's stability under field conditions and proper chemical formulations of CLA with other ingredients. However, we have demonstrated that CLA has the potential to act as a reduced-risk insecticide against an agriculturally relevant pest species.

## Supporting information

**S1 File. Greenhouse foliar spray.**
(DOCX)

## Acknowledgments

Support for this research was provided by the University of Wisconsin—Madison Office of the Vice Chancellor for Research and Graduate Education with funding from the Wisconsin Alumni Research Foundation. We would also like to acknowledge the late Mark E. Cook who was essential in the development of the ideas portrayed within this manuscript.

## Author Contributions

**Conceptualization:** Justin Clements, Russell L. Groves, Jake M. Olson.

**Data curation:** Justin Clements, Jake M. Olson.

**Formal analysis:** Justin Clements, Jake M. Olson.

**Funding acquisition:** Russell L. Groves, Jake M. Olson.

**Investigation:** Justin Clements, JoAnn Cava, Caroline C. Barry, Scott Chapman, Jake M. Olson.

**Methodology:** Justin Clements, JoAnn Cava, Scott Chapman, Jake M. Olson.

**Project administration:** Justin Clements, Jake M. Olson.

**Resources:** Justin Clements, Jake M. Olson.

**Supervision:** Jake M. Olson.

**Validation:** Justin Clements, Jake M. Olson.

**Visualization:** Jake M. Olson.

**Writing – original draft:** Justin Clements, JoAnn Cava, Jake M. Olson.

**Writing – review & editing:** Justin Clements, Russell L. Groves, Caroline C. Barry, Jake M. Olson.

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
