## [Decision Letter · Decision Letter 0]

5 Sep 2019

PONE-D-19-20291

Conjugated linoleic acid as a novel insecticide targeting the agricultural pest Leptinotarsa decemlineata

PLOS ONE

Dear Scientist Olson,

Thank you for submitting your manuscript to PLOS ONE. After careful consideration, we feel that it has merit but does not fully meet PLOS ONE’s publication criteria as it currently stands. Therefore, we invite you to submit a revised version of the manuscript that addresses the points raised during the review process.

We would appreciate receiving your revised manuscript by Oct 20 2019 11:59PM. To enhance the reproducibility of your results, we recommend that if applicable you deposit your laboratory protocols in protocols.io, where a protocol can be assigned its own identifier (DOI) such that it can be cited independently in the future. For instructions see: http://journals.plos.org/plosone/s/submission-guidelines#loc-laboratory-protocols

We look forward to receiving your revised manuscript.

Kind regards,

Yulin Gao

Academic Editor

PLOS ONE
---

## [Author Response · Author response to Decision Letter 0]

14 Oct 2019

Reviewer 1:

GENERAL COMMENTS: 

Manuscript PONE-D-19-20291 by Olson et al. reports on insecticidal and antifeedant properties of conjugated linoleic acid (CLA) against the Colorado potato beetle, Leptinotarsa decemlineata. Given the importance of this pest and its ability to develop insecticide resistance, as well as general scarcity of data on insecticidal properties of CLA, this manuscript will be valuable to a variety of applied entomologists. The reviewers general comments were as follows, “experiments appear to be properly designed and executed, although some clarification of experimental design is needed (see below). The manuscript is clearly written and is easy to read. “My one general concern is that the authors overstate the potential of CLA to become a commercial insecticide. The strength of reported negative effects on the Colorado potato beetle was moderate at best, while known biological activity in vertebrates implies possible non-target effects. Reasons for considering CLA a reduced risk, biorational, and organically certifiable insecticides were not explained. I agree that this chemical may show potential for commercialization, but the authors need to provide better support for their claims”.

SPECIFIC COMMENTS: 

Lines 27-31. Need a reference to back this up.

Response: Added citation “Alyokhin et al 2008.” 

Lines 32-34. It is not clear why the authors list these compounds. Many chemicals were used against the Colorado potato beetles since DDT.

Response: The reviewer is correct that countless insecticide chemistries have been used to control Colorado potato beetles. We have changed this sentence to read “Historically, L. decemlineata populations have been controlled with numerous insecticidal compounds including, but not limited to, DDT, paris green, arsenical pesticides, and other historical insecticides.”

Lines 44-45. What exactly are microbial and fungal synthetics? What is the difference between them and products of microbial fermentation?

Response: In this section we address some of the current investigations examining new and “safer” insecticide chemistries. Palli et al. 2014 examined RNA interference progression towards a commercial product. Šmid et al. 2015 examined the inhibition of larval growth by macrocypins, a protease inhibitor produced by parasol mushrooms. We have changed this section to read “Several biorational and reduced-risk insecticidal compounds are currently being investigated as safer alternatives for insecticidal control, including synthetic nucleic acids (RNA interference)7, bacterial and fungal secondary metabolites (macrocyprins)8, and even microbial fermentation products (spinosad)”

Line 65. “Instar larvae” is redundant. There are no early-instar pupae or adults.

Response: we have removed instar as suggested

Line 86. Which cultivar?

Response: We have clarified the tuber used as cv. Russet Burbank.

Line 87. Were potatoes grown from true seeds, or from seed tubers?

Response: Potatoes used were from seed tubers, we have clarified this in the manuscript.

Lines 98-100. Why choose that formulation? It seems that 40% is a rather high amount of impurities. Also, the listed compounds add up to 96% of the mixture. What about the remaining 4%?

Response: This particular CLA formulation was chosen as it was a commercially available, “off-the-shelf” product that is produced in reasonably large volume (tonnage production). This CLA product was considered representative of a commercial CLA that could be used for insecticide formulation. This CLA formulation results from the literature-based method of alkali isomerization of feedstock oils for generating the CLAs and other production methods have been used with limited success comparatively. We chose this formulation because the results can be rapidly transferable to industry and understood by academia. Unknowns in the formulation made up 4% of the total composition and this information was added to the manuscript.

Lines 103-104. Does “per group” mean “per well” in this context?

Response: Individual insects were placed in individual wells. We have changed the manuscript to read “Individual larvae were placed in individual wells of a 12-well Falcon (Corning Inc., Corning, New York) culture dish.” and specified the number of individuals used in each treatment group for individual experiments to clarify methods.

Lines 112, 124, and 132. It is not clear how many true statistical replications were in each of these experiments. Do n values refer to the total numbers of larvae or to the numbers of larvae per treatment? Were those arranged in groups of ten larvae per well?

Response: For each assay conducted we had a N of 50 insects that were evenly divided into 5 biological replicates of 10 insects per replicate. Each group of insects was than exposed to one of the CLA dilutions. This is described and clarified in lines 104-106.

Lines 183-186. The list of treatments is confusing. Punctuation in this sentence does not make any sense to me.

Response: To clarify treatment groups we have reworded this section to read “Each individual was placed in a petri dish and given a choice between two leaf disks (2.016 cm2); one control (acetone dipped) leaf disk and one leaf disk dipped in one of the following treatments: 1) a 16% v/v concentration of CLA in acetone, 2) a 16% v/v c9t11 methyl ester in acetone, or 3) a 16% v/v t10c12 methyl ester in acetone.”

Lines 196-199. Were those weight gains per surviving larvae, or per total larvae used in the experiment?

Response: The reviewer addresses an import question that needs to be clarified in the manuscript. Weight gain was per surviving larvae. We have clarified this in the manuscript.

Table 1. The second column is not necessary because the same colony was used for all treatments.

Response: We have removed the column entitled Population

Line 265. What is an “observed trend”? P=0.1 usually is not considered to be statistically significant.

Response: We agree with the reviewer that a P=0.1 is not generally accepted as statistically significant. In the manuscript we refrain from making the statement of an observed trend and we will more simply write, “there was a trend toward decreased…”. 

Lines 273-274. Repellence and decreased feeding activity are not the same. Larvae may be attracted to treated leaves but not feed on them due to intoxication. Experimental design chosen by the authors did not measure repellency.

Response: We have removed the terminology “repel” in this section, and have revised this statement to read, “Larvae consumed significantly greater proportions of the untreated leaf materials (acetone treatment only) in pairwise comparisons with the suspensions comprised of the following treatments: 16 % CLA v/v, 16 % c9t11 v/v, and 16% t10c12 v/v (Table 4). The parent compound CLA and its two isomers all resulted in significant reductions in feeding activity of larvae over a 24-hour period suggesting both isomers in CLA could be feeding deterrents.”.

Table 4 is missing a title.

Response: We have added a corresponding title to Table 4

Lines 274-275. What is an “active insecticide deterrent”? A chemical that deters insecticides? Is there such a thing as a passive insecticide deterrent?

Response: We have changed this sentence to read, “The parent compound CLA and its two isomers all demonstrated a significant ability to repel and decrease feeding activity of larvae over a 24-hour period suggesting both isomers of CLA could potentially function as feeding deterrents.”

Lines 286-288. What makes CLA biorational? How safe is it to non-target organisms? 

Response: The authors are suggesting that CLA could be considered as a biorational under the US EPA definition that it includes, “Pest control materials that are relatively non-toxic to people with few environmental side-effects are called “biorational” pesticides”. The potential certainly exists for CLA to have some measurable impacts upon non-target organisms that could consume potato foliage in the specific instance of this work, but it would be difficult to speculate on the potential for any non-target organism that would be present in the potato crop and not be functioning as an herbivore. 

Line 299. A 70% field mortality (in the best-case scenario) is very likely to be unacceptably low for a commercial insecticide.

Response: In the greenhouse study we observed a 70% mortality, and the reviewer is correct that this would fall beneath the desired field morality level to be considered unacceptable. However, as an original proof of concept we are observing high mortality from a minimally-formulated product. If CLA were ever to be used commercially, its formulation would need to be optimized. It was our intent to assess the insecticidal effects of a commercial- and minimally-formulated-sprayable CLA. We believe there is sufficient evidence to support optimization efforts to produce a CLA formulation that could potentially be useful for crop protection. For commercialization, a safety assessment would be recommended to assess off-target toxicity potential. The existing evidence would suggest that CLA’s biodegradation and photodegradation rates are rapid (ie. less than 4 days; EPA ECOTOX data); therefore, we suggest CLA is likely to be considered as a low-risk candidate for a biorational insecticide. 

Lines 317-318. Demonstrated biological activity in vertebrates raises serious concerns about environmental safety of CLA.

Response: We agree that bioactivity in vertebrates is a concern. However, the dosage required for bioactivity is highly unlikely to be achieved from a practical crop application. Additionally, bioactivity in animals requires repeated dosing at minimum levels of roughly 0.5% of the total diet. Since potato foliage is not readily consumed by vertebrates, we anticipate the off-target effects upon vertebrates are not likely to occur from usage on the crop. However, these and other safety concerns should be evaluated/assessed as part of a potential product development process. 

Line 341. It is too early to say whether CLA will be a highly effective insecticide.

Response: We agree and have changed the terminology to reflect the proof of concept nature of our work. We agree that rigorous advances must be made before CLA could be termed “highly effective” as an insecticide. 

Lines 345-346. Why will CLA qualify for certification as an organic insecticide? Also, what makes it a reduced-risk chemical?

Response: We have removed the term organic here as, to our knowledge, there does not exist an organic CLA designation within National Organic Program Standards, or OMRI-certification listings. However, the use of organic feedstock oil to produce CLA via the alkali isomerization process could potentially qualify the resulting CLA product as organic, provided the alkali isomerization process could qualify for organic certification. Due to the unknown nature of CLA’s organic potential, we have again, removed this statement from the discussion. Regarding the reduced risk comment, we believe CLA’s existing GRAS status for foods and specific animal feeds applications is evidence to suggest that CLA could be considered for a ‘reduced-risk’ designation from an environmental and organismal safety standpoint. However, this is not to say that appropriate toxicological assessment of CLA on aquatic and terrestrial ecosystems should be bypassed. Instead, we aim to suggest that CLA has strong potential to be useful as a reduced-risk (i.e. biorational) insecticide.

---

## [Editor Report · Decision Letter 1]

29 Oct 2019

Conjugated linoleic acid as a novel insecticide targeting the agricultural pest Leptinotarsa decemlineata

PONE-D-19-20291R1

Dear Dr. Olson,

We are pleased to inform you that your manuscript has been judged scientifically suitable for publication and will be formally accepted for publication once it complies with all outstanding technical requirements.

With kind regards,

Yulin Gao

Academic Editor

PLOS ONE

Dr. Yulin Gao

Professor

Department of Entomology

Institute of Plant Protection(IPP) 

Chinese Academy of Agricultural Sciences(CAAS) 

2# West Yuan Ming Yuan Road 

Haidian District, Beijing, 100193, P.R, China 

Office: 01062815930

Mobile: 13552643313

E-mail: gaoyulin@caas.cn 

---

## [Editor Report · Acceptance letter]

5 Nov 2019

PONE-D-19-20291R1 

Conjugated linoleic acid as a novel insecticide targeting the agricultural pest Leptinotarsa decemlineata 

Dear Dr. Olson:

I am pleased to inform you that your manuscript has been deemed suitable for publication in PLOS ONE. Congratulations! Your manuscript is now with our production department. 

With kind regards,

on behalf of

Dr. Yulin Gao 

Academic Editor

PLOS ONE